# Investigation and Comparison of Nutritional Supplement Use, Knowledge, and Attitudes in Medical and Non-Medical Students in China

**DOI:** 10.3390/nu10111810

**Published:** 2018-11-20

**Authors:** Hechun Liu, Yuexin Yang, Dengfeng Xu, Hui Xia, Da Pan, Shaokang Wang, Guiju Sun

**Affiliations:** 1Key Laboratory of Environmental Medicine and Engineering of Ministry of Education, and Department of Nutrition and Food Hygiene, School of Public Health, Southeast University, No. 87 Ding Jia Qiao Road, Nanjing 210009, China; shipinliuhechun@163.com (H.L.); withdrawxu@163.com (D.X.); 230149561@seu.edu.cn (H.X.); pantianqi92@foxmail.com (D.P.); shaokangwang@seu.edu.cn (S.W.); 2National Institute for Nutrition and Health, Chinese Centre for Disease Control and Prevention, No. 29 Nan Wei Road, Beijing 100050, China; yxyang@263.net

**Keywords:** nutritional supplements, students, university, knowledge, attitudes

## Abstract

The objective of this study is to investigate and compare the prevalence, knowledge, and attitudes of Chinese university students with respect to nutritional supplements. We conducted a cross-sectional study in several universities around China from January to December 2017, and enrolled a total of 8752 students. Of these, 4252 were medical students and 4500 were non-medical students. The use of nutritional supplements was reported by 58.9% in universities students, with a higher rate for medical students as compared to non-medical students. It was found 24.2% of participants had taken supplements in the past year. Medical students had a higher level of knowledge on nutritional supplements than non-medical students (*p* < 0.001). The most commonly used nutritional supplements were vitamin C, calcium, and vitamin B. Gender (*p* < 0.001), household income (*p* < 0.001), and health status (*p* < 0.001) were related to the nutritional supplement use after adjustment for related factors. In conclusion, in China, nutritional supplement use was found to be more common in medical students than those studying other disciplines, and was associated with sex, income, and health status. The attitude towards nutritional supplements by medical students was positive. Students’ knowledge levels about nutritional supplements need to be improved

## 1. Introduction

With improvements in the national economy and residents’ living standards, eating habits and dietary patterns have gradually changed in recent years in China [1]. An unbalanced diet may result in micronutrient deficiency [2]. In addition to nutrient intake from food, use of dietary supplements is becoming increasingly widespread [3,4], and can represent an important source of essential nutrients [5]. In both developed and developing countries, chronic non-communicable diseases (NCDs) such as cancer, type 2 diabetes, osteoarthritis, and obesity are major public health problems. Physical inactivity, alcohol use, and stress are common risk factors for these NCDs, and nutrients may alter the underlying pathologically-related mechanisms [6]. For example, antioxidants reduce oxidative damage; folic acid regulates DNA methylation; and vitamin D and calcium influence bone metabolism [7].

U.S Food and Drug Administration (FDA) has defined dietary supplements as products intended to supplement the diet and contain one or more nutritional ingredients including vitamins, minerals, herbs, botanicals, amino acids, and other substances or their constituents [8]. However, the definition of nutritional supplements (NSs) is different in China. In accordance with “Provisions on the Application and Approval for Nutritional Supplements (Notice Number 202 (2005)) [9]”, the China Food and Drug Administration (CFDA) defined nutritional supplements as products that replenish levels of vitamins and (or) minerals without providing energy, and that belong to the health food category. Their functions are to supplement the insufficiency of nutrients, prevent nutritional deficiency, and reduce the risk of certain chronic non-communicable diseases. Eight minerals, 14 vitamins, and 69 compounds are allowed to be used [10]. The health function to provide intake that is “supplemented with vitamins and minerals” [11].

Worldwide, vitamin supplements are used extensively [12]. At a recent health industry seminar in 2017, the chief of China Nutrition and Health Food reported that the value of China’s health food market had reached 400 billion yuan (USD60.2 billion) [13]. NSs, including vitamins and mineral products, account for about 35 percent of the Chinese health food market [14]. There are a variety of reasons for public supplement consumption, including the desire to fill nutrient gaps in the diet, maintain health, and reduce susceptibility to disease [15,16,17,18,19,20]. One study in the United States reported that 58% of users used dietary supplements for overall wellness and 42% used them to avoid nutritional deficiencies [21]. Another study in Australian university students found that the most common reasons were in order to be healthy and to meet nutrient needs [22]. To main good health and ensure nutrition were reported in India as the most common reasons [23]. In Malaysian study, 80% of participants stated the main reason as being “to maintain good health” [24]. However, heavy reliance on supplements is risky, as it may lead to excessive doses. For example, the Multiethnic Cohort study in the United States reported that, of people who took supplements, 50% of men and 40% of women consumed quantities that surpassed the upper limit for niacin and folate [25].

Due to the increase in NS consumption, a large number of studies on dietary supplements have been performed in the United States [26], Australia [27], Japan [28], and so on. However, supplement use in China is unclear. In America, from 1999 to 2012, the prevalence of overall dietary supplement use remained stable among adults, at around 52% [29]. In Australia, the study reported that 43.2% of adults used dietary supplements, which was higher than the rates in adolescents and children, which were 19.7% and 20.6%, respectively [30]. One study in Switzerland recruited 6188 subjects and found that the prevalence of dietary supplements use was 25.7% [31]. In some Asian countries, the prevalence was relatively lower, for example in Lebanon (23%) [32] and Jordan (27.4%) [33].

Many previous studies have paid more attention to specific groups such as the elderly, patients, and children [34,35]. However, this condition may not be the same among university students who are energetic and healthy. Due to the specific lifestyles in the campus, including stress [36], physical activity [37], eating disorders [38], and smoking and drinking [39], university students are particularly vulnerable to malnutrition, including the effects of undernutrition [40]. They seem to not depend on parents’ views in terms of use of NSs. Consumers lack sufficient knowledge on dietary supplements and do not receive basic information about interactions or side effects, and take them as healthy food without professional medical advice [41,42]. However, medical students in their future clinical practice can have positive impacts on patients health beliefs and supplement use, as well as on those of the general public [43]. Therefore, some studies also researched the dietary supplements use in health science students to find the difference, especially about their attitudes and knowledge [17,19,24,44,45]. One study conducted in the United States found that the prevalence of dietary supplements use in the students was 66.0% [46]. In Australia, one study in 2015 reported that prevalence was 56.0% among the students [47]. Another cross-sectional investigation with 1633 students found that 69% used vitamins or minerals [22]. Besides, previous studies showed the different prevalence in medical students. Almost half of American medical students in 2005 [48] and pharmaceutical students in 2017 [49] used dietary supplements. The study in India showed that 49.6% of health science students used dietary and micronutrient supplements [23]. The prevalence was lower in Croatian students (30.5%) [17] and in Dammam (29.42%) [18]. However, previous surveys enrolled participants only in one or several colleges, with a limited sample size.

With increasing NS use, many questions have arisen about characteristic of supplements in these consumers’ minds. Present studies produced results essentially in agreement with a higher prevalence of NS use in women [20,48]. In addition, there were also associations of NS use with level of education, income, and in some cases, ethnic origin [22]. Present studies have found that with higher education level and increasing age, more people used dietary supplements [48]. However, the results for income were inconsistent. One study in Australia found that the least socioeconomically disadvantaged people were more likely to use dietary supplements [27]. Besides, dietary habits, physical activities were also important factors. People who had high protein and low fat intake tended to use more dietary supplements [46]. Those with normal body mass index (BMI) who exercised regularly also used more dietary supplements compared with those who exercised little [17]. An association was also found between some chronic diseases (such as high blood pressure) and dietary supplements use in studies [27].

Due to the lack of representative national research about students and the large-scale sales of NSs, it is necessary to study the characteristics of NS users in universities and their attitudes and knowledge towards NSs. Therefore, the purpose of this study is to investigate and compare the knowledge, attitudes, and use of NS between medical students and non-medical students. The second specific research objective is to explore the association between supplement use and consumers’ sociodemographic and personal characteristics.

## 2. Methods

In China, “nutritional supplements” are defined by law, as mentioned above. Therefore, in this survey, the meaning of the term “nutritional supplements” refers to vitamin and/or minerals supplementation including forms such as health food nutritional supplements, over the counter (OTC) nutritional supplements and vitamin or mineral supplements produced in foreign countries.

Statement of Human Rights: The study was carried out in accordance with the Helsinki Declaration of 1975 as revised in 2013, and informed consent was obtained from all participants included in the study.

### 2.1. Study Design

We conducted a detailed cross-sectional questionnaire among university students in China from 1 January 2017 to 31 December 2017. The survey was divided into online and offline polls with the same questions and contents. In every university departments, tutors handed out the printed questionnaires to students and showed the websites or Qr code of online questionnaires. Each student chose only one way to finish the survey. Tutors also emphasized the basic definition of NSs. Before the formal investigation, all the participants were required to sign informed consents that were in accordance with the Declaration of Helsinki. The survey was anonymous, and the student identity number was the only identifying mark to avoid repetition. The Chinese Nutrition Society approved this study.

### 2.2. Study Participants

In this study, random cluster sampling was used to select representative cities around the seven main areas in China and then randomly choose students in medicine-related departments and other departments in local universities. The chosen universities included Jilin University (located in Jilin City, Jilin Province, in the northeast of China); Sichuan University (located in Chengdu City, Sichuan Province, in the southwest of China); Zhongshan University (located in Guangzhou City, Guangdong Province, in the south of China); Southeast University (located in Nanjing City, Jiangsu Province, in the east of China); Hebei Medical University (located in Shijiazhuang City, Hebei Province, in the north of China); Ningbo University (located in Ningbo City, Zhejiang Province, in the east of China; Jining Medical University (located in Jining City, Shandong Province, in the north of China); and Beijing Union University (located in Beijing City, in the north of China). All the participants were required to finish the questionnaire by themselves.

### 2.3. Data Collection

The students’ tutors returned completed surveys in each department. Tutors scanned and removed the incomplete questionnaires, and then all printed questionnaires were collected. Double data entry was performed using Epidata 3.1 software (EpiData Association, Odense, Denmark). After entering, the summarized data were combined with the online question data for further analysis. With respect to the online survey, if students did not finish all the questions, they could not submit it successfully. Completed data were automatically summarized and extracted.

### 2.4. Questionnaire Data

The 27-item questionnaire included four parts, which were modified by professors according to Chinese reading comprehensive habits to measure usage patterns for, attitudes about, and knowledge of vitamin and mineral products among university students in China. The first part recorded information on socio-demographic, economic, and health status. We classified ages (years) into four groups: <20, 20–24, 25–29, and ≥30 years old, respectively. Areas were divided into eight parts regarding Chinese Geographical Division [50]. We calculated body mass index (BMI) status as weight (kg) divided by the square of height (m^2^) written by participants, and grouped into underweight (<18.5), normal weight (18.5 ≤ BMI < 24.0), overweight (24.0 ≤ BMI < 28.0), and obese (≥28.0) [51]. The second part was about how they had purchased or consumed NSs. Based on these, there were some questions about the place of purchase, spending, main reasons, type, frequency, effectiveness, etc. in the second part. The third part contained six questions about basic knowledge of NSs. The fourth part aimed to determine the students’ attitude towards NSs, and included four items.

### 2.5. Pre-Survey for Reliability Test

To assess reliability of the questionnaire, we used test–retest reliability and internal consistency reliability. Cronbach’s α coefficient measured internal consistency; test–retest reliability was calculated as the intraclass correlation coefficient of two tests [52,53,54]. We gave 30 volunteers the same questionnaires twice, and the period was one month to check whether their answers were the same. The Cronbach’s α was 0.820, and the range was from 0.472 to 0.855 in different parts about knowledge, attitudes, and practice. The correlation coefficient between the two tests was 0.713, ranging from 0.333 to 0.723 in different sections. Thus, the questionnaire had good reliability, and could be applied to measure NS use. At the same time, depending on comments from respondents, wording and layout were made minor changes to ensure the clarity of the questionnaire.

### 2.6. Statistical Analysis

All data were analyzed with the statistical package SPSS, version 24.0.0. Using a two-sided test, *p* values < 0.05 were considered statistically significant. To describe the demographic characteristics, only age was the continuous variable. Thus, the mean and standard deviation were used to present the average ages and two-sample *t*-test was used for the normally distributed variable. With respect to categorical variables, the frequencies and proportions were calculated. Pearson chi squared analyses were used to compare the characteristic, practice, knowledge, and attitudes difference between medical students versus non-medical students. For multiple-choice questions, each option was treated as the variables to valuation before using multiple responses. Then frequencies and proportions were calculated. Before performing multivariate logistic regression analyses, the univariate logistic analysis was performed to examine the association of each of the demographic characteristic variables including age, sex, BMI, income, region, health status, and major, with overall NS consumption. If the *p* < 0.05, these variables were included in the further multivariate logistic regression model. Age, sex, BMI, income, region, health status, and major were all assessed as the independent variables. Multivariate logistic regression analyses were used to assess and identify factors associated with the prevalence of supplement use among students, with adjustment for all other sociodemographic variables. Odds ratio (OR) and 95% confidence intervals (CIs) were represented the results of the logistic regression analysis.

## 3. Results

### 3.1. Demographic Characteristics

A total of 8752 Chinese students participated in the study. Among the respondents, 4252 students specialized in medicine, while 4500 were non-medical students. The number of females was greater than that of males. The average age was 20.90 ± 2.52 years, and the majority of the respondents were in the age bracket of 20–24 years. Most of the students were living in the east of China, 15.5% of them lived in the north of China, and 12.5% in the southwest of China. Regarding health, 40.2% of the participants reported a good health status. Nearly 68.6% of the respondents reported that they did not have any chronic diseases, and 25.1% of them were unclear. With regard to the BMI, the majority of the respondents had a normal BMI (68.2%), while 17.7% were underweight and 14.1% were overweight or obese. Demographic characteristics and health status of participants are shown in Table 1.

### 3.2. Patterns of nutritional Supplements Consumption

24.2% of Chinese students reported that they consumed NS in the past year, which the medical students had consumed a little higher than non-medical students, *p* < 0.001. The respondents mostly bought health food nutritional supplements and OTC nutritional supplements (45.6% and 23.5%, respectively) and they preferred to buy NSs from drugstores or online. These ranks were the same in medical and non-medical students. The main reason why respondents purchased NSs was because of a doctor’s suggestion. The majority of students in both groups enrolled in our study used under RMB1000 as their expenses in terms of NS per year. Details are summarized in Table 2.

### 3.3. Prevalence of Nutritional Supplement Use

The prevalence of NS use is reported in Table 3. Out of the total students, 5098 (58.9%) had previously used NSs. However, the prevalence in medical students (61.5%) was higher than in non-medical students (56.4%), with statistical significance. Most of the users (63.4%) had consumed NSs less than one year, only 4.2% had consumed them more than five years, and there was no significant difference between the medical student group and the non-medical student group. Besides, 41.1% of the NS users had not consumed NSs in the past year; 38.2% had used NSs for less than 2 months in the past year; and only 2.9% of the users had consumed them for the whole past year. Nearly half of the users in all groups took various kinds of NSs. In addition, 43.3% of all users considered that the effectiveness of NSs was satisfactory. Nearly half of the users (49.2%) thought that the effect improved physical function.

### 3.4. Knowledge, Attitudes towards Nutritional Supplements

Table 4 and Table 5 exhibit the results of questions enrolled in the section of knowledge and attitudes. Only 22.3% of respondents gave the correct answers of the most critical function of NS. It was found that 64.0% of respondents believed that “some specific people (sick, elderly, children, pregnant women) should take NSs”. When we questioned the respondents on whether they though NSs could be used instead of a balanced diet, 83.7% of their answers were right. More than half of the respondents believed that long-term usage could lead to side effects (56.3%). Only 36.5% respondents could distinguish the difference between health food nutritional supplements and OTC nutritional supplements, while the majority of respondents could correctly select the special label for NSs in China. The one most remarkable result was that in all the questions, the percentages of medical students who answered right were all higher than those of non-medical students, with statistical significance.

Table 7 displays the association between demographic, socioeconomic factors, chronic disease, specialty and NS use. In crude model, a significantly stronger association (OR 1.47; 95%CI 1.34, 1.62) was observed between NS users and females compared with males and this association still appeared after adjustment for related factors (OR 1.37; 95%CI 1.24, 1.53). No association was observed when examining the use of NSs with age (*p* = 0.120). Neither BMI nor regions were directly associated with the use of NSs when adjusted with other factors. In terms of incomes, the prevalence of supplements use was slightly lower in higher economic status groups. NS use was associated with self-reported health status (*p* < 0.001); people who considered that their health states were excellent or poor tended to use NSs more. It was also shown that being a medical student was significantly associated with the use of NS (adjusted OR 1.21; 95%CI 1.07–1.37).

## 4. Discussion

This research investigated the prevalence of NS use in Chinese college students and their knowledge and attituded towards NSs. Moreover, we compared the differences in knowledge, attitudes, and practice status with respect to NS between medical and non-medical students, and then explored the association between NS use and influencing factors. We found that more than half of the respondents consumed NSs, with 24.2% of respondents consuming NSs in the past year. Those rates were a little higher among medical students than those among non-medical students. Also, medical students had good health knowledge on NSs. In terms of attitude towards NSs, more medical students had a favorable view compared with their similarly aged peers. After logistic regression, gender, economic and health status were the independent impact factors for NS consumption, while age, BMI, and regions were not.

To the best of our knowledge, this was the first nationwide survey to study the prevalence of NS used by in Chinese college students, and the prevalence was consistent with surveys from other countries such as 43% in Malaysia [24] and around 50% in the United States [46,47,48,49], but notably higher than in Japan [28], United Arab Emirates [19], and Iran [45], and lower than in Australia [22] and Serbia [55]. In the current study, we found that medical students tended to prefer using NSs compared to students who studied other majors; this was also found in other surveys conducted with medical and pharmaceutical students [18,56,57]. One extensive study in the United States which enrolled 33,905 participants aged between 20 and 29 years [16] found that 42% of females had used vitamin and mineral supplements in the prior month, as compared to 30% of males. Besides, about 50% of medical students had used supplements in the past month, regardless of gender [48]. In our study, most students consumed NSs because of friends’ suggestions, advertisements, or other influences rather than the doctor’s directions. Besides, most people did not consult doctors or pharmacists about their NS use. A study in Croatia reported that for students, the Internet was the most common source of information rather than healthcare professionals [17]. This problem was not found in Dammam’s research, which studied enrolled 469 male students and reported that the majority of them (65%) used supplements upon a physician’s recommendation [18]. It is important for adverse effect prevention to seek advice from physicians or pharmacists not only in China but also in other countries [58]. The other important problem is that less than half of the respondents could distinguish the difference between healthy food nutritional supplements and OTC nutritional supplements. In China, some vitamin or mineral supplements can be obtained OTC, and as such, people can get them from hospitals or chemists according to the doctor’s suggestions. There are also some health food NSs marketed in supermarkets or the Internet. Nowadays, not only some NSs are sold in markets, but also people can purchase them through the Internet [59]. Thus now, the market for health products is chaotic and the quality is not uniform in China. Additionally, the recent popular trend in NS types in China was obvious in this survey. Not only males and females but also medical and non-medical students preferred vitamin C and Ca, which were believed to be beneficial for cold and bone-disease prevention [60,61]. However, in many other studies, multivitamins were the most commonly used [22,33,47]. Early studies reported that many people reached the maximum intake of vitamins and minerals such as nicotinic acid, folic acid, and zinc [25]. One study showed that the values for nutrients, such as example vitamin E, vitamin B_12_ and zinc, might exceed the recommended intake. Excessive intake of nutrients may result in side effects and toxicity [62,63].

With respect to the knowledge level, the problem was that majority of university students did not understand the principal function of NSs; some of them believed that NSs had disease treatment and even cancer prevention functions. This problem was not apparent in developed countries. For example, students in Australia believed in taking NSs in order to meet nutrient needs, not to treat diseases [22]. Besides, NSs were related to side effects such as gastrointestinal and neurological complications, toxic hepatic injury, and drug interactions [64,65,66]. In our study, nearly half of respondents thought overdosing or consuming NSs for long time periods had no toxic effects. It was reported that NSs can cause adverse effects in children and old people, and adverse effects most frequently occurred in the 20–34 year age group [67]. Other research also reported that side effects linked to these NSs occurred among the younger population [59]. In our study, more medical students correctly answered each question about NSs as compared to non-medical students. A study in India found that compared to nursing and dental students, medical students scored the highest percentage in their supplement knowledge [23]. However, some studies [56,57] still reported that the knowledge about NSs was poor among medical or pharmaceutical students. This might be because they only had a general view of the NSs, but no knowledge about the clinical characteristics, such as the product efficacy, supplemental dosage, and adverse reactions. Hence, it is necessary for medical or pharmacy students to have more knowledge about NSs and develop a scientific view about NS use, so that they can use NSs correctly and give to their patients adequate advice in the future [28]. A study in Serbia found that students educated in pharmacology mastered adverse reactions of dietary supplements better than students without this education [44]. Therefore, it is essential to comprehensively understand NSs, but in China, medical students were insufficiently educated about NSs which was consistent with other studies [68,69]. One research in Missouri reported that more than half of pharmacists received questions about nutrition products from patients, but only 2.4% could answer them correctly [70]. In another study, general medical practitioners knew a little about side effects and adverse effects of dietary and herbal supplements but did not regularly discuss them with patients [71]. Another huge problem is the interaction between NSs and prescription drugs. A study using National Health and Nutrition Examination Survey (NHANES) data found that 34.3% of American adults and 47.3% of patients diagnosed with chronic disease took prescription drugs and dietary supplements together [72]. These studies clearly emphasized the necessity for medical staff to master the drug–supplement interactions well, so as to identify adverse interactions and provide adequate suggestions to patients [49].

Regarding the association between demographic characteristics and NS users, most of our findings were the same with previous studies [26,62,73,74,75]. Many previous literatures have shown that female, old and highly educated were more likely to use NSs [22,45,55]. In our study, the participants were all college students and the age range was not large, so there were no significant differences according to the age and education levels. An early study showed that females had greater health awareness [76], so they may pay more attention to NSs and try to use them more. Besides, early studies reported that people of a higher economic position were more likely to take supplements [62,77], because they took more care of their health and had more disposable income, which may motivate them to consume more [74]. However, our study did not reflect this. The reason might be that all participants were students, and the health educational level was nearly the same. Previous studies showed different results related to BMI and its association with supplement intake; however, there was no relationship between BMI and supplement use in the current study. Students with normal BMI took significantly more supplements in a study in Croatia [17]. One study in Taiwan found that there was no relationship between BMI and vitamin and mineral supplements such as calcium and vitamin E. Similarly [78], NHANES, conducted in 2008 and focusing on the use of supplements, did not find any association between BMI and supplement consumption [79]. The present study did not report the association between chronic diseases and supplement use; however, previous studies reported that there was an association between self-reported health and supplements use [73]. People who suffered from chronic disease more tended to take supplements in the same studies, but other studies found opposite conclusions. For instance, a study in the United Kingdom found that people with high blood pressure were less likely to use NSs than healthy people [73]. A study in Australia found the opposite result [27]. The contradiction might be due to variations among populations under study [47], as our study enrolled mostly young people, and a few of them suffered from chronic diseases, thus their health awareness was poor.

The strength of the present study was the wide-range data, which had the certain representativeness of NS use in Chinese students. There were also some limitations to the study. First, this survey was not conducted in every province in China and did not investigate degrees and specific majors such as arts, business, and engineering. Further study should explore the difference between NS use according to the degrees and majors. Second, the medical students and non-medical students did not match well with statistical results, which showed that there were differences in gender, age, etc. Furthermore, some Chinese consumers, even college students, did not understand the difference between NSs and dietary supplements. A few people may be confused about the definition, and they may think fish oil and proteins are also NSs. Indeed, here the term “NSs” only refers to vitamin and mineral supplements in the questionnaire. Lastly, the questionnaire lacked a dietary survey part and physical activity survey part to find the association between the health behavior and NSs. People who take supplements are more likely to have better health behaviors and engage in exercise. It is also more likely for them to meet the recommendations for fruits and vegetable intake [26,62,73].

## 5. Conclusions

Our study shows that the use of NSs was more common among medical students than non-medical students in universities in China, and the total prevalence of NS use was around half among the Chinese students, with the main types being vitamin C, calcium, and vitamin B. Besides, we found that, although the knowledge level about NS among medical students was higher than non-medical students, they still needed to be improved. Moreover, the attitudes of medical students towards NSs were more positive. Finally, gender as well as economic and health status were associated with NS use. Further research should focus on the relationship between different lifestyles (including exercise and eating habits) and supplement use in order to understand the drivers of supplement use, which will be helpful to guide Chinese students in order to use NSs appropriately.

## Figures and Tables

**Table 1 nutrients-10-01810-t001:** Demographic characteristics of Chinese students in the survey.

Characteristic		All Subjects *n* = 8752	Medical Students *n* = 4252 (48.6)	Non-Medical Students *n* = 4500 (51.4)	*p*-Value
Sex, *n* (%)	Male	3185 (43.7)	1178 (37.9)	2007 (48.0)	<0.001
	Female	4103 (56.3)	1930 (62.1)	2173 (52.0)	
Age, years, *n* (%)	Average	20.90 ± 2.52	21.21 ± 2.60	20.60 ± 2.42	<0.001
	<20	2250 (26.2)	991 (23.6)	1259 (28.7)	<0.001
	20–24	5736 (66.8)	2754 (65.7)	2982 (67.9)	
	25–29	555 (6.5)	429 (10.2)	126 (2.9)	
	≥30	45 (0.5)	18 (0.4)	27 (0.6)	
BMI status, *n* (%)	Underweight	1518 (17.7)	718 (17.2)	800 (18.1)	0.469
	Normal	5859 (68.2)	2874 (68.8)	2985 (67.7)	
	Overweight/obese	1212 (14.1)	587 (14.0)	625 (14.2)	
Annual per capita income, Yuan (RMB), *n* (%)	<5000	1972 (23.5)	933 (22.3)	1039 (24.6)	<0.001
	5000–9999	1856 (22.1)	990 (23.7)	866 (20.5)	
	10,000–29,999	2021 (24.0)	1070 (25.6)	951 (22.5)	
	30,000–49,999	982 (11.7)	504 (12.0)	478 (11.3)	
	50,000–99,999	981 (11.7)	458 (10.9)	523 (12.4)	
	>100,000	594 (7.1)	228 (5.5)	336 (8.7)	
Area of residence, *n* (%)	Northeast China	623 (7.3)	523 (12.6)	100 (2.3)	<0.001
	Northern China	1312 (15.5)	245 (5.9)	1067 (24.6)	
	Central China	737 (8.7)	140 (3.4)	597 (13.8)	
	Eastern China	3847 (45.3)	2492 (59.9)	1355 (31.3)	
	Southwest China	1059 (12.5)	196 (4.7)	863 (19.9)	
	Southern China	449 (5.3)	338 (8.1)	111 (2.6)	
	Northwest China	450 (5.3)	222 (5.3)	228 (5.3)	
	Overseas	13 (0.2)	5 (0.1)	8 (0.2)	
Self-reported health status, *n* (%)	Excellent	1696 (19.5)	807 (19.0)	889 (19.9)	<0.001
	Very good	3500 (40.2)	1810 (42.6)	1690 (37.8)	
	Good	2958 (34.0)	1413 (33.3)	1545 (34.6)	
	Fair	362 (4.2)	160 (3.8)	202 (4.5)	
	Poor	75 (0.9)	16 (0.4)	59 (1.3)	
	Unclear	120 (1.4)	39 (0.9)	81 (1.8)	
Chronic disease, *n* (%)	Absent	5980 (68.6)	3132 (73.8)	2848 (63.6)	<0.001
	Present	552 (6.3)	234 (5.5)	317 (7.1)	
	Unclear	2188 (25.1)	877 (20.7)	1311 (29.3)	

mean ± standard deviation; frequency (percentage). BMI body mass index. Chi-squared test.

**Table 2 nutrients-10-01810-t002:** Nutritional supplements purchasing behavior analysis.

Variables	All Subjects, *n* = 8752	Medical Students *n* = 4252 (48.6)	Non-Medical Students *n* = 4500 (51.4)	*p*-Value
Have you consumed any nutritional supplements in the past year?				
Yes	2102 (24.2)	1179 (27.8)	923 (20.7)	<0.001
No	6600 (75.8)	3068 (72.2)	3532 (79.3)	
Which type of nutritional supplements have purchased the most in the past year?				
Health food	939 (45.6)	575 (48.3)	364 (41.8)	0.001
Over the counter (OTC)	485 (23.5)	283 (23.8)	202 (23.2)	
Products from foreign countries	229 (11.1)	128 (10.8)	101 (11.6)	
Other	143 (6.9)	80 (6.7)	63 (7.2)	
Not clear	265 (12.9)	124 (10.4)	141 (16.2)	
Where do you buy nutritional supplements?				
Drugstore	1113 (55.2)	641 (55.0)	472 (55.4)	0.159
Hospital	185 (9.2)	119 (10.2)	66 (7.7)	
Internet	317 (15.7)	180 (15.5)	137 (16.1)	
Direct selling	131 (6.5)	78 (6.7)	53 (6.2)	
Overseas	129 (6.4)	77 (6.6)	52 (6.1)	
Other	142 (7.0)	70 (6.0)	72 (8.5)	
What was the main reason for your first purchase?				
Doctor’s suggestion	605 (30.2)	378 (32.7)	227 (26.7)	<0.001
Friends’ suggestion	515 (25.7)	287 (24.8)	228 (26.8)	
Reference, related books	293 (14.6)	203 (17.6)	90 (10.6)	
Advertisements	90 (4.5)	40 (3.5)	50 (5.9)	
Direct selling	49 (2.4)	29 (2.5)	20 (2.4)	
Seller’s suggestion	62 (3.1)	25 (2.2)	37 (4.3)	
Other	392 (19.5)	193 (16.7)	199 (23.4)	
How much do you spend on supplements per year? Yuan (RMB)				
>10,000	59 (2.9)	15 (1.3)	44 (5.1)	<0.001
5000–9999	60 (3.0)	21 (1.8)	39 (4.5)	
1000–4999	189 (9.3)	106 (9.1)	83 (9.6)	
<1000	1716 (84.8)	1019 (87.8)	697 (80.8)	

mean ± standard deviation; frequency (percentage). Chi-squared test.

**Table 3 nutrients-10-01810-t003:** Nutritional supplements consumption behavior analysis.

Variables	All Subjects *n* = 8752	Medical Students *n* = 4252 (48.6)	Non-Medical Students *n* = 4500 (51.4)	*p*-Value
Have you taken any nutritional supplements?				
Yes	5098 (58.9)	2606 (61.5)	2492 (56.4)	<0.001
No	3555 (41.1)	1630 (38.5)	1925 (43.6)	
For how long have you taken nutritional supplements?				
Less than 1 year	3163 (63.4)	1655 (63.8)	1508 (63.1)	0.931
1–2 years	1118 (22.4)	581 (22.4)	537 (22.5)	
3–5 years	496 (9.9)	257 (9.9)	239 (10.0)	
More than 5 years	209 (4.2)	103 (3.9)	106 (4.4)	
Duration of consumption over the past year:				
None	2072 (41.9)	1065 (41.3)	1007 (42.5)	0.204
≤2 months	1891 (38.2)	1029 (39.9)	862 (36.4)	
3–5 months	646 (13.0)	301 (11.7)	345 (14.6)	
≥6 months	200 (4.0)	93 (3.6)	107 (4.5)	
A whole year	142 (2.9)	93 (3.6)	49 (2.1)	
What kind of nutritional supplements do you take?				
Single	2200 (45.2)	1130 (44.4)	1070 (46.2)	0.204
Various	2664 (54.8)	1417 (55.6)	1247 (53.8)	
How is the effectiveness of your nutritional supplements?				
Full	370 (7.7)	222 (8.8)	148 (6.4)	<0.001
Fair	2086 (43.3)	1132 (44.9)	954 (41.5)	
Neutral	891 (18.5)	416 (16.5)	475 (20.7)	
Not sure	723 (30.6)	750 (29.8)	723 (31.4)	
What is the main effectiveness of nutritional supplements do you think?				
To improve immunity	861 (25.2)	419 (23.8)	442 (26.6)	0.034
To improve body function	1683 (49.2)	883 (50.2)	800 (48.1)	
To treat diseases	322 (9.4)	184 (10.5)	138 (8.3)	
Other	555 (16.2)	273 (15.5)	282 (17.0)	

mean ± standard deviation; frequency (percentage). Chi-squared test.

**Table 4 nutrients-10-01810-t004:** Knowledge on nutritional supplements by Chinese students.

Variables	Number (%) That Answered Correctly			*p*-Value
	All Subjects	Medical Students	Non-Medical Students	
What is the most important function of nutritional supplements?	1862 (22.3)	1061 (25.9)	801 (18.8)	<0.001
Do you think some specific people should take nutritional supplements?	5553 (64.0)	2967 (69.9)	2586 (58.4)	<0.001
Do you know nutritional supplements cannot replace a balanced diet?	7257 (83.7)	3629 (85.6)	3628 (81.9)	<0.001
Do you think overdosing on nutritional supplements has toxic effects?	4883 (56.3)	2565 (60.5)	2318 (52.3)	<0.001
Do you know how to distinguish between health food supplements, and OTC supplements?	3119 (36.5)	1873 (44.5)	1246 (28.7)	<0.001
Can you correctly select a special label for nutritional supplements in China?	6336 (75.7)	3226 (77.5)	3110 (73.9)	<0.001

When the respondents were asked whether they thought it was necessary to take NSs, they responded “No” at 70.1%. A handful of respondents considered that they would encourage those in the neighboring environment to take supplements (35.2%). The study participants enrolled considered “tablets” as preferred form of NS as Table 5.

**Table 5 nutrients-10-01810-t005:** Attitude towards nutritional supplements.

Variables	All Subjects	Medical Students	Non-Medical Students	*p*-Value
Do you think it is necessary to take nutritional supplements?				
Yes	2568 (29.9)	1308 (31.4)	1260 (28.5)	0.01
No	6017 (70.1)	2861 (68.6)	3156 (71.5)	
Why you think it is necessary?				
To avoid dietary deficiency	1459 (48.6)	834 (49.2)	625 (47.9)	0.096
To prevent diseases	459 (15.3)	259 (15.3)	200 (15.3)	
To promote recovery	126 (4.2)	78 (4.6)	48 (3.7)	
To treat diseases	58 (1.9)	31 (1.8)	27 (2.1)	
To increase health	731 (24.4)	386 (22.8)	345 (26.4)	
Other	167 (5.6)	106 (6.3)	61 (4.7)	
Would you encourage your friends to take it?				
Yes	3051 (35.2)	1698 (40.0)	1353 (30.5)	<0.001
No	5622 (64.8)	2545 (60.0)	3077 (69.4)	
Which form of supplements do you like?				
Tablets	3327 (38.9)	1738 (41.3)	1589 (36.5)	<0.001
Capsules	1914 (22.4)	931 (22.1)	983 (22.6)	
Liquids	2343 (27.4)	1116 (26.5)	1227 (28.2)	
Others	973 (11.4)	419 (10.0)	554 (12.7)	

The most commonly used NSs were vitamin C (23.0%), followed by calcium (20.2%), vitamin B (12.1%). These ranks were the same in all groups, whether divided by major or gender (Table 6).

**Table 6 nutrients-10-01810-t006:** Types of nutritional supplements used by medicine and sex *.

	Total	Students		Sex	
	*n* (%)	Medical *n* (%)	Non-Medical *n* (%)	Males *n* (%)	Females *n* (%)
Folic Acid	301 (2.5)	157 (2.6)	144 (2.3)	124 (3.2)	109 (1.8)
Vitamin B	1477 (12.1)	705 (11.7)	772 (12.6)	461 (11.9)	745 (12.5)
Vitamin E	1192 (9.8)	659 (10.9)	533 (8.7)	327 (8.5)	627 (10.5)
Vitamin C	2799 (23.0)	1434 (23.8)	1365 (22.3)	818 (21.2)	1458 (24.5)
Vitamin D	887 (7.3)	461 (7.6)	426 (6.9)	329 (8.5)	400 (6.7)
Se	234 (1.9)	129 (2.1)	105 (1.7)	101 (2.6)	92 (1.5)
Ca	2453 (20.2)	1249 (20.7)	1204 (19.6)	764 (19.8)	1220 (20.5)
Fe	546 (4.5)	263 (4.4)	283 (4.6)	169 (4.4)	248 (4.2)
I	214 (1.8)	124 (2.1)	90 (1.5)	95 (2.5)	84 (1.4)
Multi-vitamins	890 (7.3)	383 (6.4)	507 (8.3)	289 (7.5)	437 (7.3)
Multi-mineral	313 (2.6)	120 (2.0)	193 (3.1)	115 (3.0)	149 (2.5)
Multi-vitamin and mineral	484 (4.0)	180 (3.0)	304 (5.0)	160 (4.1)	235 (3.9)
Others	367 (3.0)	163 (2.7)	204 (3.3)	108 (2.8)	156 (2.6)

* Respondents may have reported using more than one supplement.

**Table 7 nutrients-10-01810-t007:** Demographic and lifestyle characteristics of study participants using nutritional supplements.

Characteristics	User %	Unadjusted		Adjusted	
		OR (95% CI)	*p*-Value	OR (95% CI)	*p*-Value
Sex					
Males	1650 (39.6)	1		1	
Female	2514 (60.4)	1.47 (1.34, 1.62)	<0.001	1.37 (1.24,1.53)	<0.001
Age (years)			0.014		0.120
<20	1300 (26.0)	0.83 (0.45, 1.55)	0.568	0.53 (0.20, 1.39)	0.195
20–24	3319 (66.3)	1		1	
25–29	362 (7.2)	0.84 (0.45, 1.57)	0.580	0.60 (0.23, 1.58)	0.304
≥30	27 (0.5)	1.12 (0.59, 2.14)	0.723	0.56 (0.21, 1.50)	0.249
BMI			0.014		0.950
<18.4	930 (18.6)	1.05 (0.93, 1.19)	0.452	0.99 (0.85, 1.16)	0.893
18.5–23.9	3388 (67.7)	1		1	
>24.0	685 (13.7)	1.23 (1.05, 1.43)	0.009	1.01 (0.84, 1.23)	0.908
Household incomes, Yuan (RMB)			<0.001		<0.001
<5000	1069 (21.8)	1		1	
5000–9999	1012 (20.7)	0.75 (0.62, 0.91)	0.003	0.69 (0.55, 0.87)	0.001
10,000–29,999	1197 (24.5)	0.76 (0.62, 0.91)	0.004	0.69 (0.55, 0.87)	0.002
30,000–49,999	639 (13.1)	0.92 (0.76, 1.11)	0.388	0.85 (0.68, 1.06)	0.154
50,000–99,999	619 (12.6)	1.20 (0.97, 1.49)	0.088	1.15 (0.89, 1.49)	0.282
>100,000	358 (7.3)	1.08 (0.87, 1.33)	0.503	1.06 (0.82, 1.36)	0.669
Region			<0.001		<0.001
Northeast China	424 (8.6)	0.41 (0.11, 1.49)	0.174	0.63 (0.11 3.52)	0.600
Northern China	750 (15.2)	0.65 (0.18, 2.37)	0.510	1.04 (0.18, 5.93)	0.961
Central China	417 (8.4)	0.42 (0.12, 1.54)	0.192	0.68 (0.12, 3.80)	0.658
Eastern China	2202 (44.5)	1		1	
Southwest China	600 (12.1)	0.40 (0.11, 1.47)	0.169	0.65 (0.12, 3.67)	0.626
Southern China	289 (5.8)	0.40 (0.11, 1.47)	0.167	0.76 (0.14, 4.27)	0.755
Northwest China	258 (5.2)	0.56 (0.15, 2.06)	0.383	0.85 (0.15, 4.81)	0.850
Overseas	10 (0.2)	0.41 (0.11, 1.52)	0.182	0.66 (0.12, 3.76)	0.644
Self–reported health status			<0.001		<0.001
Excellent	818 (16.1)	2.61 (1.79,3.80)	<0.001	2.77 (1.74, 4.42)	<0.001
Very good	2081 (41.0)	1.44 (0.99, 2.11)	0.059	1.57 (0.98, 2.52)	0.063
Good	1860 (36.6)	1		1	
Fair	232 (4.6)	2.29 (1.57, 3.33)	<0.001	2.36 (1.48, 3.77)	<0.001
Poor	38 (0.7)	2.78 (1.81, 4.27)	<0.001	2.60 (1.54, 4.39)	<0.001
Unclear	47 (0.9)	1.55 (0.87, 2.78)	0.140	1.48 (0.74, 2.96)	0.265
Chronic disease			<0.001		<0.001
Absent	3360 (66.2)	1		1	
Present	402 (7.9)	0.85 (0.76, 0.93)	<0.001	0.93 (0.82, 1.05)	0.215
Unclear	1316 (25.9)	1.82 (1.47, 2.25)	<0.001	1.97 (1.53, 2.55)	<0.001
Major					
Non-medical	2492 (48.9)	1		1	
Medical	2606 (51.1)	1.23 (1.13,1.35)	<0.001	1.21 (1.07, 1.37)	0.003

Adjusted OR and *p*-values were based on multiple logistic regression analysis, adjusted for all other characteristics included in the table.

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
