# Peer review of "Investigation and Comparison of Nutritional Supplement Use, Knowledge, and Attitudes in Medical and Non-Medical Students in China"

_nutrients, 2018, doi:10.3390/nu10111810_

Reviewer 1 Report

It´s recomended improve the introduction section about the reason for the supplement consumption.

In methods section have to indicate the ethical committe and informed consent for students.

In the samply, Is there other health science students? It is suggested to perform an analysis for differente degrees or type of students.

Author Response

Response to Reviewer 1 Comments

Dear Reviewers:

Thank you for your comments concerning our manuscript entitles “Investigations and comparison of nutrients supplement use, knowledge and attitudes between medical and non-medical students in China” (ID: 380294). Those comments are very helpful for revising and improving our paper. We have studied comments carefully and have made correction, which we hope meet with approval. Here are the point-to-point responses and the lines numbers were presented in the final pattern for the review not other mark patterns for the review in Microsoft:

Point 1: It´s recomended improve the introduction section about the reason for the supplement consumption.

Response 1: We are very sorry for our negligence of reasons for the consumption. After reviewing and summering the previous researches, the main reasons for supplement consumption were to fill nutrient gaps in the diet, maintain health and reduce susceptibility to disease. The details were presented in the line 63 to 70, which we added in the introduction part.

Point 2: In methods section have to indicate the ethical committe and informed consent for students.

Response 2: It is really true as Reviewer suggested about ethical committee. Please let us explain why there was no ethical committee. Our research was a cross-sectional study. And we only gave participants questionnaires to get the information. We did not use any human tissues or bloods. Besides, we did not give any interventions. Hence, in our university, this study did not need to be approved by the Ethics Committee. But all subjects gave their informed consent before they participated in our study. We have added the explanation details about informed consent in line 257-258 in the method part. I hope it will not affect our further reviews. 

Point 3: In the samply, Is there other health science students? It is suggested to perform an analysis for differente degrees or type of students.

Response 3: We are very sorry for our unclear explanation about medical students. In our study, the medical students mean students who will get the medical degree when graduated. They were in the clinical medical school, basic medical school, public health school and pharmacy school. Besides, it was really useful as reviewer suggested preforming an analysis for different degrees or type of students. It was our limitation there was no statistic about degrees and the specific majors of students. However, other majors were not our focus and purposes. We only get the information about whether their major are related to the medicine. We have added this limitation in the discussions in the line 361 to 363. And we will pay high attention to these aspects in future researches.  

Special thanks to you for these good comments again. 

Reviewer 2 Report

1. Better care with typing should be taken in the paper. Many clerical errors throughout should be fixed.

2. I would suggest the authors add a strong concluding statement at the end of the abstract.

3. Introduction: Some statements in par 1are made without any supporting reference.

4.   Introduction: The authors didn't include some important studies on the topic and I would suggest to referring to. For example: (Nutrition 2016; 32(5): 524-530), (J Clin Diagn Res 2014; 8(8): HC10-HC13), (Int J Environ Res Public Health 2018; 15(6): 1058).  

5. Introduction: This section fails to build a good rationale or to show novelty in conducting the present study. No studies from other context were used to tightly support the research objective. The literature should be in more depth and discuss the association between NS use and demographic, socioeconomic factors and chronic disease.

6. Methods: I'm not convinced this is an appropriate study design (Line 94-105). There is definitely not enough information on the study design, survey data collection and methods for reliability testing.

7. Methods: The data analysis is very week in its current form: The data analysis needs to be described sufficiently in this manuscript in order to interpret the findings.

8. Results: Clear rationale why males, age 20-24 years, BMI (18.5-23.9) and East China were used as a reference category?

9. Discussion: The supporting literature in this section could benefit from a more recent scan of the literature.

Author Response

Response to Reviewer 2 Comments

Dear Reviewers:

Thanks for your comments concerning our manuscript entitles “Investigations and comparison of nutrients supplement use, knowledge and attitudes between medical and non-medical students in China” (ID: 380294). Those comments are all valuable and have the important guiding significance to our researches. Based on your comment and request, we have made extensive modification on the original manuscript. Here, we attached revised manuscript. Revised portion are marked in the manuscript. The main responses to the reviewer’s comments are as following and the lines numbers in manuscript were presented in the final pattern for the review not other mark patterns for the review in Microsoft.

Point 1: Better care with typing should be taken in the paper. Many clerical errors throughout should be fixed.

Response 1: we are very sorry for our incorrect writing. We revised the manuscript carefully to minimize grammatical, and bibliographical errors. The changes we marked were in red in revised paper.

Point 2:  I would suggest the authors add a strong concluding statement at the end of the abstract.

Response 2: we are very sorry for our negligence of conclusion in the abstract. We have added it “in China, the nutrients supplements use is higher in medical students than others and the usages are associated with sex, income and health status. Students’ knowledge level about nutrients supplements need to be improved and the attitude towards nutrients supplements of medical students is more positive.” in the abstract in line 23 to 26.

Point 3: Introduction: Some statements in par 1are made without any supporting reference.

Response 3: As reviewer reminded, we checked it carefully and added the corresponding references. For example, in line 31, we added the references about dietary change trends to suppose “eating habits and dietary patterns change gradually in recent years in China”. And in line 33, we added the references about dietary supplements use increase to suppose “other dietary supplements forms are becoming increasingly widespread”. More changes were marked in the paper.

Point 4: Introduction: The authors didn't include some important studies on the topic and I would suggest to referring to. For example: (Nutrition 2016; 32(5): 524-530), (J Clin Diagn Res2014; 8(8): HC10-HC13), (Int J Environ Res Public Health 2018; 15(6): 1058).  

Response 4: It is really true as Reviewer suggested important studies. These articles were really useful. We read them carefully and searched other related articles. Introduction and discussion parts enriched the references. For example,

Article in (Nutrition 2016; 32(5): 524-530) was about dietary supplements use in an Australian university students, which we referred in line 58, 88, 97 and 336. Articles in (J Clin Diagn Res2014; 8(8): HC10-HC13) was also about KAP of dietary supplements in health science students, which we referred in line 59,91 and 315. Articles in (Int J Environ Res Public Health 2018; 15(6): 1058) was the recent research about supplements in Croatia, which we referred in line 55, 84 and 347. More added reference were presented in the paper.

Point 5: Introduction: This section fails to build a good rationale or to show novelty in conducting the present study. No studies from other context were used to tightly support the research objective. The literature should be in more depth and discuss the association between NS use and demographic, socioeconomic factors and chronic disease.

Response 5: we have re-written the introduction part according to the reviewers’ suggestions. We presented the background in line 33-63 and research gaps in 64-93; we summarized the related present studies in line 53-93. One of our objectives is to investigate and compare the knowledge, attitude and practice status of NS between medical students and non-medical students. We summarized the previous studies to support this objective in line 73-93. The second objective is to explore the association between supplements use and consumers’ sociodemographic, and personal characteristic. We summarized the previous studies to support this objective in line 94-105. The discussion about association between NS use and demographic, socioeconomic factors were added in line 94-105.

Point 6: Methods: I'm not convinced this is an appropriate study design (Line 94-105). There is definitely not enough information on the study design, survey data collection and methods for reliability testing.

Response 6: We have added the study design in line 118-126 “A descriptive cross-sectional questionnaire was conducted among university students in China from January 1st 2017 to December 31st 2017. The survey was divided in the forms of online and offline questionnaires with the same questions and contents. In every university departments, tutors handed out the printed questionnaires to students and showed the websites or Qr code of online questionnaires. Each student chose only one way to finish the questionnaire. Tutors also emphasized on the basic definition about NS. Before the formal investigation, all the participants were required to sign the informed consents that were accordance with the Declaration of Helsinki. The investigation was anonymous, but the student ID was the only identifying mark in order to avoid repetition. The Chinese Nutrition Society approved this study”.

Besides, we added the data collection in line 141 to 146. “Completed surveys were returned by the students’ tutors in each department. Tutors scanned and removed the incompletely questionnaires, then all printed questionnaires were collected. Double data entries were performed using Epidata 3.1 software. After entering, the summarized data combined with the online question data for the further analysis. About online survey, if students did not finish all the questions, they cannot submit successfully and completed data can be automatically summarized and extracted.”

Finally, we described the reliability testing details in line 162-171. “To assess reliability of the questionnaire, test-retest reliability and internal consistency reliability were used. Internal consistency was measured by Cronbach’s α coefficient; test-retest reliability were calculated as intraclass correlation coefficient of two tests[52-54]. We gave 30 volunteers the same questionnaires twice, and the period time was 1 month to check whether their answers were the same. The Cronbach’s α was 0.820, and the range was from 0.472 to 0.855 in different parts about knowledge, attitudes, and practice. Besides, the correlation coefficient between two tests was 0.713 ranged from 0.333 to 0.723 in different parts. So the questionnaire had good reliability, which could be applied to measure the NS use. At the same time, depended on comments from respondents, wording and layout were made minor changes to ensure the clarity of the questionnaire.”

Point 7: Methods: The data analysis is very week in its current form: The data analysis needs to be described sufficiently in this manuscript in order to interpret the findings.

Response 7: We have improved the data analysis descriptions about two-sample t-test, Pearson chi square analyses, multiple response, logistic regression analyses and univariate logistic analysis in line 173-189. Actually, before the logistic regression analysis, we did the univariate analyses to find the impact factors. However, because of space limit, these results were not presented in the manuscript. If possible, we could add them as the supplement materials.

Point 8: Results: Clear rationale why males, age 20-24 years, BMI (18.5-23.9) and East China were used as a reference category?

Response 8: about how to choose the dummy variables is the logistic regression analysis. One way is to choose the first category, which is convenient for further comparisons. The other way is to choose the most large sample size category. Because of the large sample, the model has the better robustness. Whether using males or females as reference category, it will not have the significant influence of the results because of the only two categories. Using 20-24 as the reference category, because the participants in this group were most and the proportion of nutrients supplements use in this group was higher than other groups. BMI (18.5-23.9) is the normal value. And the BMI (18.5-23.9) group was the largest. We wanted to explore compared with the normal BMI people, whether underweight or obesity people will use more or less dietary supplements. So we used BMI (18.5-23.9) as the reference categories. About East China, because the number of participants in this group is the most and traditionally the east areas is the most developed region of China. We also wanted to explore whether the related poor regions will influence the students consumption.

Point 9: Discussion: The supporting literature in this section could benefit from a more recent scan of the literature.

Response 9: Considering the reviewer’s suggestion, we improved the discussion part supported by more recent literature. For example, we referred more important studies in 283 -287, and so on. We made the discussion more depth and coherent. The changes were all marked in the manuscript.

We tried our best to improve the manuscript and we hoped meet with approval. Special thanks to you for these good comments again. 

Round  2

Reviewer 2 Report

The authors have done a good job revising the paper. However, the paper needs to be edited by a native English speaker throughout to adhere to conventions of written English. There are still clerical/grammatical errors throughout the paper.

Author Response

Dear reviewer,

Thanks for your kind comments on our paper entitled "Investigation and Comparison of nutrients supplements use, knowledge and attitude between medical and non-medical students in China".

According to the comments, we have improved the English writing by a native English-speaking colleague who helped us to check through the paper and modified the grammatical and clerical errors. 

Here, we attached revised manuscript. Revised portion are marked in the manuscript.

Thank you again for your review and valuable suggestions.
